# A Combined Linkage and GWAS Analysis Identifies QTLs Linked to Soybean Seed Protein and Oil Content

**DOI:** 10.3390/ijms20235915

**Published:** 2019-11-25

**Authors:** Tengfei Zhang, Tingting Wu, Liwei Wang, Bingjun Jiang, Caixin Zhen, Shan Yuan, Wensheng Hou, Cunxiang Wu, Tianfu Han, Shi Sun

**Affiliations:** 1Ministry of Agriculture and Rural Affairs Key Laboratory of Soybean Biology, Institute of Crop Sciences, Chinese Academy of Agricultural Sciences, Beijing 100081, China; wutingting@caas.cn (T.W.); lwwmaize@163.com (L.W.); 18331121822@163.com (C.Z.); yuanshan@caas.cn (S.Y.); houwensheng@caas.cn (W.H.); wucunxiang@caas.cn (C.W.); hantianfu@caas.cn (T.H.); 2National Center for Transgenic Research in Plants, Institute of Crop Sciences, Chinese Academy of Agricultural Sciences, Beijing 100081, China

**Keywords:** soybean, protein content, oil content, quantitative trait loci (QTL), linkage analysis, genome-wide association study (GWAS), candidate genes

## Abstract

Soybean is an excellent source of vegetable protein and edible oil. Understanding the genetic basis of protein and oil content will improve the breeding programs for soybean. Linkage analysis and genome-wide association study (GWAS) tools were combined to detect quantitative trait loci (QTL) that are associated with protein and oil content in soybean. Three hundred and eight recombinant inbred lines (RILs) containing 3454 single nucleotide polymorphism (SNP) markers and 200 soybean accessions, including 94,462 SNPs and indels, were applied to identify QTL intervals and significant SNP loci. Intervals on chromosomes 1, 15, and 20 were correlated with both traits, and QTL *qPro15-1*, *qPro20-1*, and *qOil5-1* reproducibly correlated with large phenotypic variations. SNP loci on chromosome 20 that overlapped with *qPro20-1* were reproducibly connected to both traits by GWAS (*p* < 10^−4^). Twenty-five candidate genes with putative roles in protein and/or oil metabolisms within two regions (*qPro15-1*, *qPro20-1*) were identified, and eight of these genes showed differential expressions in parent lines during late reproductive growth stages, consistent with a role in controlling protein and oil content. The new well-defined QTL should significantly improve molecular breeding programs, and the identified candidate genes may help elucidate the mechanisms of protein and oil biosynthesis.

## 1. Introduction

With an average composition of approximately 40% protein and 20% oil, soybean (*Glycine max* (L.) Merr.) is the most important source of vegetable protein and edible oil, accounting for 68% of total global protein consumption [1] and more than half of global oilseed production [2]. Breeders have the goal of producing soybean varieties with high-protein and oil content, traits that are quantitatively controlled by multiple genes that have small effects and are significantly influenced by the environment [3,4,5]. A strong negative correlation between protein and oil content has been verified in previous studies [6,7], suggesting that some quantitative trait loci (QTL) may inversely affect protein and oil content. Identifying and studying QTL associated with protein or oil content is important for directing molecular breeding, and identifying genes and gene functions that affect protein and oil content. 

To find genetic markers that are near genes controlling traits of interest, linkage analysis can be performed using biparental segregating populations [8]. Since Diers et al. [3] first used linkage analysis to discover a major QTL connected to soybean protein and oil content on chromosome (Chr.) 20, studies have been conducted to detect QTL near various types of markers, including amplified fragment length polymorphism (AFLP) markers, restriction fragment length polymorphism (RFLP) markers, and simple sequence repeat (SSR) markers in biparental populations [7,9,10,11,12,13]. The Soybase website has listed 255 and 322 QTL linked to protein and oil content, respectively, involving every chromosome in the biparental population (http://soybase.org/, 8 July 2019). However, the limited overlap of protein/oil-interrelated markers and sparse density of molecular markers used in previous reports have inhibited the identification of candidate genes within the wide QTL intervals and limited the increase in protein or oil content resulting from marker-assisted selection (MAS) [14]. Requirements for the construction of secondary mapping populations and the use of map-based cloning have slowed down application to breeding. Using recently developed high-density single nucleotide polymorphism (SNP) markers based on high-throughput sequencing, Seo et al. [2] identified 23 protein and oil QTL within small regions that covered 14 linkage groups using 1570 SNP markers, including *qHPO20*, a QTL significant for both protein and oil content that overlapped a previously reported QTL [13,15,16,17]. Wang et al. [18] constructed two high-density genetic maps that contained 4000 more SNP markers, examined loci related to soybean evolutionary traits, and predicted candidate genes that related to these traits. Patil et al. used a high-resolution bin map (3343 SNP markers) to detect 18 QTL connected to soybean seed protein, oil, and sucrose content QTL that were then confirmed by a genome-wide association study (GWAS) [5].

GWAS, based on linkage disequilibrium (LD), is a prevailing strategy to find genetic variations that affect complex traits by using genome-wide markers combined with phenotypes [8,19]. Hansen et al. [20] first successfully applied GWAS to plant genetics, tightly linking the *B* gene to the annual growth habit of sea beet using genome-wide AFLP markers. In recent years, GWAS has been applied to analyze complex quantitative traits in soybean such as protein, oil, fatty acid, and amino acid content and salinity tolerance in the different wild, landrace, and elite soybean lines, yielding putative candidate genes based on bioinformatic analysis in order to identify their action mechanisms [21,22,23]. Lee et al. [24] gathered 621 soybean accessions from maturity group I–IV and 34014 SNP markers to identify QTL for protein, oil, and amino acid content. They also detected some QTL on Chr. 5, 10, 15, and 20 that coincided with previous results [3,25,26,27].

Compared with linkage analysis, association analysis does not require the construction of a mapping population and can analyze multiple alleles from the same locus simultaneously [19]. Due to abundant recombination accumulated during the long-term evolution of natural populations, the results are of higher resolution that can even be located within individual genes [8,28]. However, population structure and genetic relationships may lead to false positive results in association analyses [29]; hence, it is best to combine linkage analysis and association analysis for the most accurate QTL results. Combined analysis methods have successfully mapped loci to associated traits in rice [30,31] and maize [32,33], but it is rarely employed to study soybean protein and oil content. In this study, we combine linkage analysis and GWAS methods to detect QTL and identify candidate genes that are linked to protein and oil content.

## 2. Results

### 2.1. Phenotypic Variation of Protein and Oil Contents in Two Panels

Three hundred and eight recombinant inbred lines (RILs) and 203 soybean accessions were used in this study. The seed protein and oil content of two panels grown over three years in three different locations are summarized in Table 1. The protein content in the two parent lines for the RILs, Zigongdongdou (ZGDD) and Heihe27 (HH27), averaged over different locations, was 44.55% and 40.81%, respectively, and the average oil content was 18.69% and 20.07%, respectively. Differences in protein and oil content between the parent lines were significant in Sanya in 2016, 2017, 2018 (16SY, 17SY, 18SY), and Xinxiang in 2018 (18XX) (*p* < 0.01), but not in Xiangtan in 2017 (17XT). Data for the parent lines grown in Xinxiang in 2017 (17XX) were not available. The mean content for RILs was between the parents, with transgressive segregation expanding the range. The skewness and kurtosis indicate that the data conforms to a normal distribution that is apparent in the histograms in Figure 1, suggesting that both protein and oil content are controlled by multiple genes that can be analyzed by linkage analysis. The protein and oil content in the association panel also follow a normal distribution that is conducive to GWAS and have a wide phenotypic variation for traits as indicated by the variance, range, and coefficient of variance (CV) observed (Table 1).

Variance analysis (ANOVA) (Table 1) revealed that significant differences (*p* < 0.01) were found in genotype, environment, and genotype × environment interactions for the two traits. Broad-sense heritability (*H*^2^) of both traits was high (0.83~0.90), demonstrating that genetic factors play a vital role in the accumulation of protein and oil in these lines.

### 2.2. Genetic Map and QTL Analysis of Protein and Oil Contents

Seven thousand one hundred and twenty-three SNP markers were filtered to construct a genetic map. Markers with severe segregation distortion (*x*^2^ > 100) were removed through Joinmap 4.1. The final map included 3454 SNP markers covering 20 linkage groups (LGs) that spanned 2208.16 cM of the genome with an average distance of 0.64 cM between adjacent markers. There was an average of 173 SNP markers in each LG, ranging from 70 (on Chr. 11) to 260 (on Chr. 3) [34].

Using the genetic map, we identified QTL that were co-detected by two algorithms: inclusive composite interval mapping (ICIM) and a mixed model based on composite interval mapping (MCIM) and/or consistently detected in multiple environments, and combined QTL that exist in two adjacent intervals as the same QTL. This resulted in the identification of seven protein content QTL and eight oil content QTL that were located on 11 chromosomes (Table 2 and Figure 2). The limit of detection (LOD) value (the threshold for ICIM) of these QTL ranged from 2.90 to 35.35 while the F values (the threshold for MCIM) were from 4.80 to 26.20, and these QTL explained 1.56% to 23.98% of the phenotypic variation. The QTL with positive values for the additive effect indicates that the ZGDD parent contributes to the allele that is conducive to the trait. The QTL on Chr. 1, 15, and 20 are linked to both protein and oil content. Among those QTL, *qPro15-1*/*qOil15-1* contributed to a high phenotypic variation explanation (PVE) (13.40%~17.81%), was localized to a narrow physical region (from 2691560 bp to 3476238 bp), and was indicted by both algorithms and three different environments. Therefore, this QTL interval was further examined to identify candidate genes. A second, oil-related QTL *qOil5-1*, was also detected by two algorithms, apparent in every environment, and contributed a large PVE ranging from 7.04% to 23.98%. *qPro15-1*, *qPro20-1*, and *qOil5-1* were significant QTL intervals identified in at least three environments, having high LOD/F value and contributing more than 7% PVE (Table 2 and Appendix A).

### 2.3. Genome-Wide Association Study (GWAS) Results

Two hundred and three soybean accessions consisting of a diverse range of protein and oil content were genotyped, yielding 3,977,183 SNPs and 491,910 indels. After filtering for missing rates ≤ 10%, minor allele frequencies ≥ 5% and LD pruning, 94,462 SNPs and indels were available for GWAS.

Principal component (PC) analysis was conducted with 94,462 SNPs and indels and three outlier cultivars (Hai 94, Wuhuasiyuehuang, and Suidaohuang) were identified and removed from the association panel. The first three PCs dominate the population structure (Figure 3a), they divide the population into two main groups which exhibit a geographic distribution pattern (Figure 3b). The first subgroup primarily consisted of cultivars from the northeast region of China (NER) and the USA, while the second subgroup mainly included cultivars from the Huang-Huai region of China (HHR) and the south region of China (SR). However, a few accessions from NER and USA (e.g., liaodou15 and Hood) were sorted into the second subgroup and a few accessions from HHR and SR (e.g., qihuang10 and taiwan75) were placed in the first subgroup (Figure 3b,c), perhaps due to their parents’ origin area. Population structure analysis indicated *K* = 2 was the modeling choice (Figure 3c and Appendix A), and the result was confirmed by the PC analysis. The heat map of the population shows their kinship that can distribute into two subpopulations (Figure 3d). The physical distance of total LD decay, where *r*^2^ dropped below 0.1, was approximately 132 kb (Figure 3e), and we also detected the LD decay of accessions from NER, USA, HHR, and SR with the relative LD decay distances of 180, 190, 171, and 161kb, respectively.

To minimize false positives due to population structure, we performed a general linear model (GLM) and a mixed linear model (MLM) and found that MLM effectively reduced false positive SNPs. A threshold of –log (*P*) = 4 was determined as the criteria for detecting significant signals of protein and oil content. Further, we conducted GWAS on two sub-population panels (the NER-USA and HHR-SR sub-populations). A total of 19, 12, and 36 SNP loci distributed on 17 chromosomes were detected in the NER-USA and HHR-SR sub-populations and total population, respectively (Appendix A and Appendix A). The *p*-values of all significant SNP loci were from 9.37 × 10^−7^ to 9.90 × 10^−5^. One SNP on Chr. 5, 9, 13, 16, 18, and three SNPs on Chr. 20 (41133383, 35512580, and 34990940) were associated with both traits in one environment, and these significant SNP loci on Chr. 20 were associated with both protein and oil content in 17XX and 18XX.

### 2.4. Co-Detected Results by Linkage Analysis and GWAS

We combined the results of linkage analysis and GWAS to identify SNP regions that were co-detected by both analyses (Table 3).

Four significant SNP loci regions distributed on Chr. 2, 6, 9 and 20 were co-detected, and all the SNP loci detected by GWAS were distributed in the QTL intervals obtained by linkage analysis. The co-detected SNP regions on Chr. 2 and Chr. 6 had a weak PVE (<5%), but likely included genes that exerted modest effects. The region on Chr. 20 was linked to both protein and oil content, with a higher PVE for protein content (7.24~9.39%).

### 2.5. Candidate Genes and Expression Levels

We next searched for candidate genes in a co-detected SNP region from Chr. 20 and an extra strongly indicated QTL interval from Chr. 15. Based on the LD decay distance of total population and four regions, we extended the regions about 200 kb that from 34.60 to 35.40 Mb including the SNP loci on Chr. 20. The physical region of *qPro15-1*/*qOil15-1* on Chr. 15 was from 2.60 to 3.50 Mb. We focused on genes that were indicated by annotation information to be involved in protein or oil metabolism as candidate genes. Nine and 16 genes were selected on Chr. 15 and Chr. 20, respectively, that were predicted to have one of these four categories of function: structural components, metabolic enzymes, material transporters, and regulators of gene expression (Table 4).

Quantitative real-time PCR (qRT-PCR) was applied to measure the relative expression of the 25 candidate genes, identifying eight genes that had significant differences in the expression levels between the two parent lines at late reproductive growth stages in pods. *Glyma.15g033200*, *Glyma.15g034100*, *Glyma.20g105300*, *Glyma.20g106900,* and *Glyma.20g107600* shared a pattern of low expression levels with no significant difference between the two parents up to 25 days after the R3 period, then the expression level of these genes in ZGDD increased sharply compared to HH27 (Figure 4a). As shown in Figure 4b, the expression of *Glyma.15g034600* and *Glyma.20g103200* was higher in HH27, but generally low in both parents until 25 days after the R3 period. Then, 30 days after the R3 period, gene expression switched to be significantly higher in ZGDD only to reverse again by the 35th day to be highly expressed in HH27 relative to ZGDD. *Glyma.15g040100* has its own distinct pattern. For the first 10 days after the R3 period, both parents had high expression levels, then the expression decreased with expression in HH27 being significantly higher than in ZGDD (Figure 4c). The detailed expression patterns for the other five genes are shown in Appendix A.

## 3. Discussion

### 3.1. The Accuracy of QTL Analysis and GWAS is Improved by Using Phenotypic Data from Different Locations and Employing Ample SNP Markers

The two soybean populations we studied were planted in three different geographical locations. Previous studies have shown that soybean varieties originating from higher latitudes possess lower protein content and higher oil content and that the protein content of the same soybean variety is negatively correlated with latitude, altitude, day length, and the oil content, while being positively correlated with temperature and moisture [7,35,36]. Song et al. [37] also found that crude protein content was positively correlated with accumulated temperature ≥ 15 ℃ and mean daily temperature. In our study, the protein and oil content of ZGDD/HH27 were different when grown in three different locations, especially in XT (Table 1); the comprehensive climate factors in XT must have led to the significant change in protein and oil content of HH27, producing a wide phenotypic variation in the RILs used for QTL analysis. Growth in multiple environments also expanded the phenotypic variation of the association panel (Table 1). The use of plants grown over multiple years at different locations helped us reduce environmental factors to identify genes that consistently affect these traits in our QTL and GWAS analyses.

Increasing the number of markers also improves the accuracy of QTL analysis and GWAS. The application of AFLP, RFLP, and SSR markers in QTL localization of soybean protein and oil content has previously been limited, resulting in imprecise QTL region identification [29,38]. In this study, 3454 SNP markers obtained by simplified genome resequencing were used to increase the resolution of the genetic map (0.64 cM of average distance between adjacent markers) and reduce the physical interval of the QTL (average distance was about 1.5 Mb). For GWAS, LD decay distance determines the minimum saturation marker density and using more markers produces a higher probability of detecting functional sites [19]. Nearly one hundred thousand filtered SNP and indel markers were used in this study, improving the precision of GWAS to study the complex traits.

### 3.2. Refined QTL Intervals and SNP Loci for Protein and Oil Content Were Identified

Although multiple protein and oil content QTL have been previously discovered, few have been effectively used in breeding due to their small phenotypic effects and poor reproducibility, so identifying QTL with consistent, large effects is desirable [7,39]. In our linkage analysis study, QTL that were detected by both ICIM and MCIM algorithms and/or stably detected in multiple growth environments were identified. All of the QTL intervals overlapped or were close to regions reported by previous studies (Table 2). Here, we followed up on the QTL located on Chr. 15 and 20 that were significantly and consistently correlated to both protein and oil content. Diers et al. [3] located an oil-related QTL near the RFLP marker Pb on Chr. 15, which was close to the physical region of *qPro15-1*/*qOil15-1* identified in this study and a similar region was further associated with soybean protein or oil content in other QTL studies [2,12,26]. Some SNP loci correlated to fatty acid and amino acid phenotypes were also included in this region [15,22], but candidate genes were not discovered. A second QTL, *qPro20-1*/*qOil20-1* identified in our study, also overlapped with a previously reported QTL region. Reinprecht et al. [40] detected a protein and oil QTL adjacent to the marker Satt270 which included a protein content QTL identified by Lu et al. [41] and an oil content QTL found by Qi et al. [16]. Using high-density SNP markers obtained by genome resequencing, Patil et al. [5] also detected a protein content QTL in the physical region from 33.8 to 37.4 Mb on Chr. 20 in two environments. Our refined QTL will make MAS breeding more accurate and efficient. As to an oil QTL on Chr. 5, GWAS studies have shown that some SNP loci in this region regulated oil content and hence they searched for the candidate genes [29,42]. Zhang et al. [22] associated a SNP locus at Chr. 5: 41883826 bp with oil content, and identified a candidate gene *Glyma.05g245000* that annotated as 3-Oxo-5-asteroid-4-dehydrogenase. Lee et al. [24] discovered five significant oil-related SNP loci positioned within 41.75~41.89 Mb on Chr. 5, and they listed some previous QTL for protein and oil content that were in the same region identified by our linkage analysis results. This supports the reliability of our results and suggests that there must be some oil regulating genes in this region.

In the GWAS results, the SNP loci on Chr. 20 interested us because they were embedded in the QTL intervals that had high PVE. Priolli et al. [43] detected a SNP locus that associated with fatty acid components near marker Satt270 on Chr. 20 and that is located in the SNP loci region identified in this study. However, since other GWAS studies have identified a different region of Chr. 20 from 29 to 34 Mb [5,15,24,29,42], this study has likely discovered a novel region to excavate for candidate genes.

### 3.3. The Candidate Genes Differentially Expressed at Late Reproductive Growth Stage between Both Parents Will Be Further Analyzed 

Comparing the results identified as QTL intervals and SNP loci, we decided the co-detected SNP loci region 34.60~35.40 Mb extended by approximately 200 kb on Chr. 20, and the *qPro15-1*/*qOil15-1* interval 2.60~3.50 Mb on Chr. 5, an extra region, were the best novel regions for candidate genes, rather novel QTL intervals, that might control protein and/or oil metabolism. The synthesis and catabolism of protein and oil are complex biochemical processes [44,45] and we looked for the genes that might play roles in protein and/or oil metabolism based on annotated information. 

To test the 25 genes predicted to influence the accumulation of protein and oil content, we looked for differential relative expression levels during R3 to R8 growth stages in the two parents ZGDD and HH27, resulting in the identification of eight candidate genes for further study. Previous studies have shown that the accumulation of protein and oil content is most concentrated during the late reproductive growth stage and they are negatively correlated at this stage [46,47]. The energy needed to produce oil in seed mainly comes from saccharides and protein, and some varieties of protein can be degraded into acetyl-CoA, which is the raw material of oil [46,48]. Protein and oil accumulate in the developing seeds of pods, but the surrounding pods can transport matter into the seeds, so we extracted RNA from the whole pods. Here, we discovered that eight genes had significantly different expression levels at a late reproductive growth stage between the two parents, ZGDD derived from low latitude of China that was grown in short-day conditions (12 h light/12 h dark) and HH27 derived from high latitude of China that was grown in long-day conditions (16 h light/8 h dark). Among these eight genes, five genes of *Glyma.15g033200* (structural constituent of ribosome), *Glyma.15g034100* (acyltransferase activity), *Glyma.20g105300* (serine/threonine kinase family protein), *Glyma.20g106900* (translation initiation factor 3 (IF-3) family protein), and *Glyma.20g107600* (phospholipase-like protein) had similar expression patterns (Figure 4a), the significantly up-regulated expression of these genes in ZGDD at the late reproductive growth stage might be the reason for high protein content of ZGDD. The significantly up-regulated expression of two genes *Glyma.15g034600* (drug transmembrane transport, transport of citric acid, and malic acid) and *Glyma.20g103200* (tryptophan, anthranilate synthase) in HH27 at the late reproductive growth stage (Figure 4b) might contribute to the high oil content of HH27. *Glyma.15g040100* (ACT domain-containing protein, metabolic process like protein synthesis and degradation), expressed stably but significantly higher in HH27 (Figure 4c), could also have created the higher oil content in HH27. These results provide preliminary evidence for the possible roles of these genes played in the accumulation of protein and/or oil content. However, environmental conditions have a great influence on protein and oil content. Since the parents were grown with two different photoperiod treatments, a possible role for photoperiod will be addressed in follow up experiments to determine whether gene expression and accumulation of protein and oil content are related to photoperiod or the variety itself. Different protein and oil content between cultivars from diverse regions may be correlated to variations in photoperiods.

## 4. Materials and Methods

### 4.1. Plant Materials and Field Trials

The plant material included a linkage panel and an association panel. The linkage panel consisted of RILs from 308 F_2:7_ lines derived from a cross between HH27 (protein content is 39%, oil content is 21%) and ZGDD (protein content is 45%, oil content is 19%). RILs and their parents were grown in six environments: Sanya (SY, 18°23′N, 109°11′E), Hainan province in 2016, 2017, and 2018; Xiangtan (XT, 27°47′N, 112°55′E), Hunan province in 2017; and Xinxiang (XX, 35°18′N, 113°55′E), Henan province in 2017 and 2018. The association panel, composed of 203 soybean accessions that included 182 accessions from China (94 from NER, 50 from HHR, 38 from SR) and 21 accessions from the USA (Appendix A), was planted in 18SY, 17XT, 17XX, and 18XX. Both panels were grown in a randomized complete block design with two replications. The arrangement was 1.5 m long rows with 0.5 m row spacing and 0.1 m of distance between individuals.

### 4.2. Phenotypic Data and Analysis 

Fourier transform-near infrared reflectance (FT-NIR) spectrometry (Bruker, Karlsruhe, German) was applied to scan the near infrared absorption spectra of the dry seeds. Under the Quant 2 method of OPUS v. 4.2 software (Bruker, Karlsruhe, German), the samples’ protein and oil content data were calculated using the dry basis model [49]. Each RIL and soybean accession from each replication of each environment was detected three times using about 150~200 dry seeds per detection, with the average used in statistical analysis. The histogram of phenotypic data was constructed using EXCEL (Microsoft, Redmond, WA, USA). Statistical analysis of phenotypic data and ANOVA was conducted using SAS v. 9.4 (SAS Institute, Cary, NC, USA), with type III analysis being employed. The *H*^2^ of protein and oil contents was calculated using the following equation [50]:$$H^{2}=\sigma_{G}^{2}/(\sigma_{G}^{2}+\sigma_{G*E}^{2}/e+\sigma_{e}^{2}/re$$
in which $\sigma_{G}^{2}$ is genetic variance, $\sigma_{G*E}^{2}$ is genotype × environment variance, $\sigma_{e}^{2}$ is error variance, *r* is the number of replications, and *e* is the number of environments.

### 4.3. Genotyping and Linkage Analysis

For RILs, 2b-RAD technology [51] was applied to do simplified genome resequencing. Qualified libraries were paired-end sequenced on the Illumina Hiseq Xten platform to obtain high-quality SNP markers widely distributed throughout the genome. Using Joinmap v. 4.1 [52] to construct the genetic map, markers beyond the LOD threshold 5.0 were scattered into 20 LGs [34]. A regression algorithm and the Kosambi function were used to calculate the map distances (in cM) between adjacent markers.

QTL were predicted using two software packages based on different algorithms. IciMapping v. 4.1 [53] used the ICIM algorithm, with the following default parameters: mapping method was ICIM-ADD, step was 1 cM, PIN was 0.001, and LOD threshold was manual input of 2.5. QTLNetwork v. 2.1 [54] employed the MCIM algorithm, with the permutation performed 1000 times, the F-threshold set to 4.7, and testing window and walk speed set to 10 and 1 cM, respectively.

### 4.4. Genotyping and GWAS

The 203 soybean cultivars were sequenced with the high-throughput next-generation sequencing platform of Hi-Seq 2000 with an average sequencing depth of 10-fold and genotyped using the Genome Analysis Toolkit (GATK) pipeline [55]. After being trimmed by TRIMMOMATIC (parameter: illuminaclip: adaptor. seq: 2:30:10 trailing: 3 sliding window: 4:10 MINLEN: 20), clean reads were mapped to the soybean reference genome (v. Wm82.a2.v1) by BWA [56] with default parameters, SNPs/InDels were called by GATK (-stand_call_conf set to 30.0, -stand_emit_conf set to 10.0, and -glm set to BOTH). The variations were then recalibrated by a Gaussian mixture model, and outliers were discarded. Variants were further filtered by BCFtools (v. 1.2, QUAL ≥ 50.0, DP ≥ 5.0, QD ≥ 5.0, MQ ≥ 30, MAF ≥ 0.03, Coverage ≥ 90%). InDels longer than 6 bp were discarded. More than four million SNPs and indels were obtained. SNPs and indels were filtered with missing rates ≤ 10% and minor allele frequencies ≥ 5% using PLINK [57]. The sequencing data of 125 accessions used in this study have been deposited into the NCBI database under Short Read Archive (SRA) accession number SRP062560, and the sequencing data of the rest 78 accessions used only in this study have been deposited into SRA database in NCBI under accession number PRJNA589345. Linkage disequilibrium value was calculated using the LD composite method in SNPRelate software and highly-linked SNPs were pruned, and LD plots were modified via locally weighted scatterplot smoothing (LOWESS) using R software and testing smoothing parameters fixed to 0.01 [58]. PCA was conducted by SNPRelate, and 3 cultivars of population bias were removed based on the PCA result. The software fastSTRUCTURE was used to analyze the population structure (*K* = 2, 3, 4, 5) [59] and it was verified based on the PCA result. Association signals of seed protein and oil were identified based on 94,462 SNPs and indels from 200 samples with MLM by the first three PCs and kinship in GAPIT [60]. The LD analysis was calculated by using the squared allele frequency correlation (*r*^2^) in PopLDdecay [61]. The critical threshold was set as *p* < 10^−4^ to declare the significant SNP loci in GWAS. 

### 4.5. Identification and Verification of Candidate Genes 

Based on the LD decay distance, 200 kb upstream and downstream of regions near the significant SNP loci on Chr. 20 were explored to identify genes whose functional annotation related to the metabolism of protein and/or oil in the soybean reference genome Williams 82 (http://www.soybase.org/). The functional annotation was from TAIR (www.arabidopsis.org/), GO (http://geneontology.org/), PFAM (http://pfam.xfam.org/), PANTHER(http://www.pantherdb.org/) databases and KOG (clusters of orthologous groups for eukaryotic complete genomes) annotation. Similarly, the QTL interval on Chr. 15 were also scanned to identify candidate genes. qRT-PCR was applied to identify the relative expression of candidate genes in the pods of two parents: ZGDD planted in a short-day greenhouse (12 h light/12 h dark) and HH27 planted in a long-day greenhouse (16 h light/8 h dark) to simulate their suitable light conditions in order to get protein and oil content close to that in the originate region (the parent ZGDD originates from low latitude area of China (Zigong, Sichuan province, 29°20′N, 104°46′E), it is sensitive to photoperiods and can only blossom and mature in a short-day condition; the parent HH27 derives from high latitude area of China (Heihe, Heilongjiang province, 50°14′N, 127°31′E), it is insensitive to photoperiods and can blossom and mature in both long-day and short-day conditions.). Pods were picked from the middle nodes of the main stem every five days from the R3 through the R8 stage, with three replicates for each plant [62]. The entire pods were used for the isolation of total RNA using TransZol Up (Transegen Biotech, Beijing, China). First-strand cDNA was synthesized from 1 µg of the total RNA using a FastQuant RT Kit (Tiangen Biotech, Beijing, China). For qRT-PCR, 10 µL reaction volume was applied using KAPA SYBR^®^ FAST qPCR Kits (KAPA Biosystems, Wilmington, MA, USA) with the following components: 1 µL of 1:5 diluted cDNA, 0.2 µL of each primer (10 µM), 5 µL of 2 × SYBR FAST qPCR Master Mix, 0.2 µL of 50 × ROX Low Reference Dye, and water to a final volume of 10 µL. QuantStudio 7 Flex (Applied Biosystems, Waltham, MA, USA) was used to run the qRT-PCR with following conditions: hold stage was 95 °C for 3 min; PCR stage was 40 cycles of 95 °C for 5 s and 60 °C for 30 s; melt curve stage was 95 °C for 15 s, 60 °C for 1 min and 95 °C for 15 s. All PCR reactions were run in triplicate. Data were analyzed using the 2^−ΔΔCt^ method with the mRNA level of the GmActin (*Glyma.18g290800*) gene used as the internal control. The primers used are shown in Appendix A. 

## 5. Conclusions

In summary, using linkage analysis and GWAS, we detected 15 reproducible and significant QTL intervals and 67 significant SNP loci that affect the protein and/or oil content of soybeans. We searched the co-detected SNP region on Chr. 20 and an extra QTL interval on Chr. 15 to identify 25 candidate genes that may regulate the accumulation of soybean protein and oil. Among them, eight genes had differential expression patterns in the parent lines (ZGDD and HH27) at late reproductive growth stages. Further experiments with these gene candidates should lead to a better understanding of the molecular mechanisms of protein and oil biosynthesis in soybean. The refined QTL intervals and SNP loci in our study could also improve molecular breeding based on these markers.

## Figures and Tables

**Figure 1 ijms-20-05915-f001:**
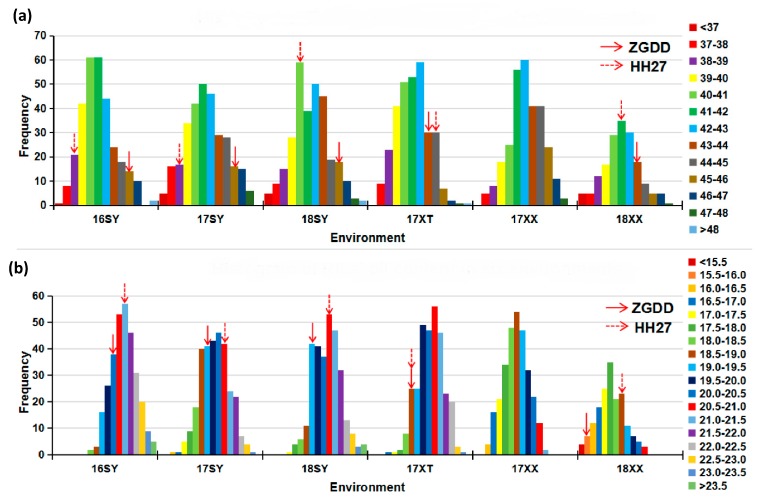
Histogram of recombinant inbred lines (RILs)’ protein content (**a**) and oil content (**b**) in six environments. 16SY, 17SY, 18SY, 17XT, 17XX, 18XX represent different environments of Sanya, Xiangtan, Xinxiang in 2016, 2017, and 2018. Zigongdongdou (ZGDD) and Heihe27 (HH27) are the parents of the RILs. Bars in different colors represent different content of protein/oil.

**Figure 2 ijms-20-05915-f002:**
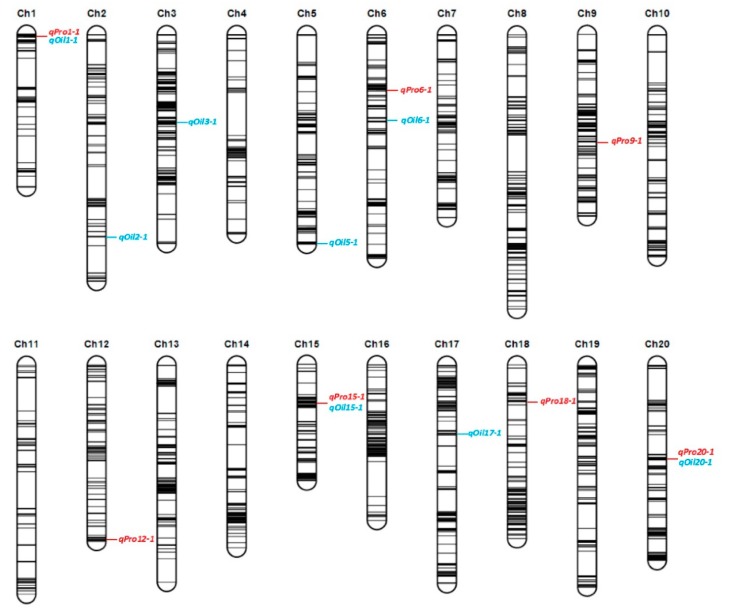
Location of quantitative trait loci (QTL) related to protein and oil contents. QTL in red color were protein while in blue were oil.

**Figure 3 ijms-20-05915-f003:**
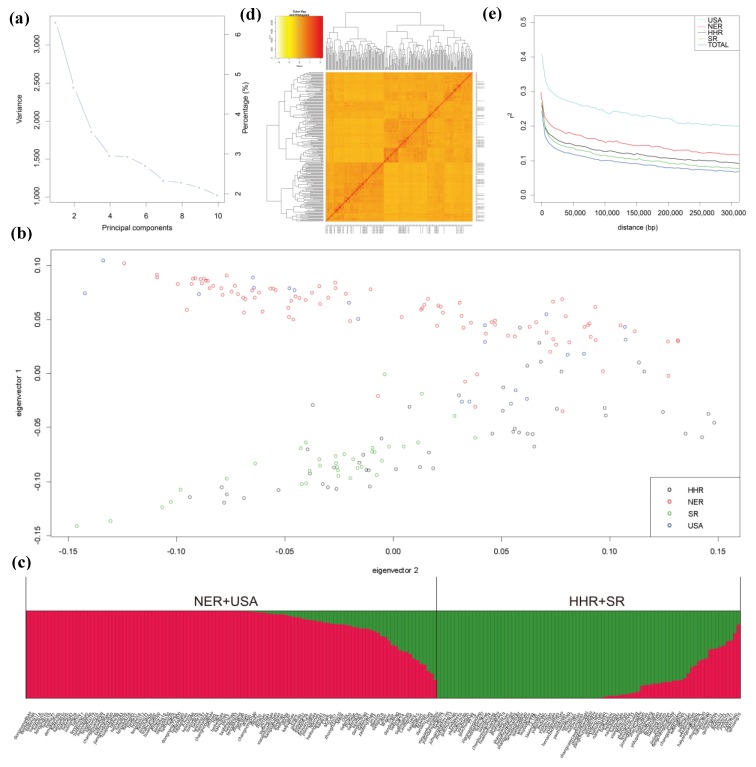
The principal component (PC) analysis (**a**,**b**), population structure analysis (**c**), heat map of the kinship matrix of the 203 soybean accessions (**d**), and linkage disequilibrium (LD) decay (**e**) of the association panel.

**Figure 4 ijms-20-05915-f004:**
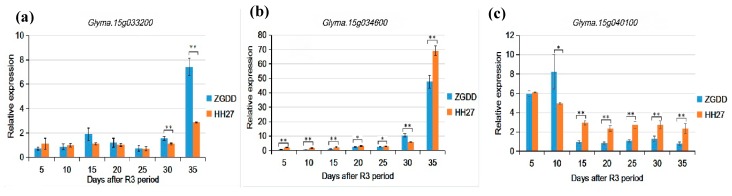
Relative expression patterns of candidate genes. *Glyma.15g033200*, *Glyma.15g034100*, *Glyma.20g105300*, *Glyma.20g106900*, and *Glyma.20g107600* express as (**a**), *Glyma.15g034600* and *Glyma.20g103200* as (**b**), *Glyma.15g040100* as (**c**). * *p* < 0.05, ** *p* < 0.01.

**Table 1 ijms-20-05915-t001:** Descriptive statistics and variance analysis for protein and oil content of two panels in multiple environments.

Population	Trait	Environment ^a^	Parents		Means (%)	Variance	Range (%)	CV ^c^ (%)	Skewness	Kurtosis	F Value of Variance Analysis			*H* ^2^ ^d^
			HH27 ^b^ (%)	ZGDD ^b^ (%)							Genotype (G)	Environment (E)	G*E	
RILs ^g^	Protein	16SY	38.55	45.33	41.54 ± 0.12	4.52	36.29~48.64	5.12	0.49	0.11	14.33 ***^,f^	47.63 ***	2.75 ***	0.83
		17SY	38.94	45.17	41.89 ± 0.14	6.33	35.90~48.00	6.00	0.15	−0.46				
		18SY	40.15	45.01	41.93 ± 0.14	5.65	33.48~48.50	5.67	0.07	0.20				
		17XT	44.65	43.68	41.58 ± 0.11	4.03	37.35~48.38	4.83	0.16	−0.17				
		17XX	NA ^e^	NA	42.62 ± 0.12	4.14	37.53~47.29	4.77	−0.10	−0.34				
		18XX	41.74	43.57	41.48 ± 0.17	5.14	33.19~47.23	5.46	−0.18	0.72				
	Oil	16SY	21.40	20.21	21.13 ± 0.06	1.17	18.13~24.38	5.11	0.08	−0.08	18.27 ***	1533.15 ***	2.57 ***	0.87
		17SY	20.53	19.13	19.93 ± 0.07	1.47	16.12~23.17	6.09	0.00	−0.18				
		18SY	20.73	19.43	20.54 ± 0.07	1.37	17.49~24.26	5.69	0.16	0.04				
		17XT	18.83	18.91	20.38 ± 0.06	1.18	16.76~23.14	5.32	−0.20	−0.14				
		17XX	NA	NA	18.71 ± 0.06	1.12	16.19~21.44	5.65	−0.04	−0.49				
		18XX	18.86	15.77	17.81 ± 0.09	1.39	15.13~20.90	6.62	0.17	−0.09				
Accessions	Protein	18SY	-	-	42.02 ± 0.23	12.42	33.40~51.33	8.39	0.19	−0.45	20.28 ***	3.94 **	3.15 ***	0.86
		17XT	-	-	42.21 ± 0.17	7.08	35.51~49.22	6.30	−0.01	−0.33				
		17XX	-	-	42.10 ± 0.20	8.83	36.10~48.83	7.06	0.22	−0.66				
		18XX	-	-	42.46 ± 0.21	8.58	30.68~49.84	6.90	−0.51	1.01				
	Oil	18SY	-	-	20.73 ± 0.11	2.84	15.65~23.94	8.12	−0.54	−0.33	27.54 ***	535.37 ***	2.95 ***	0.90
		17XT	-	-	21.14 ± 0.09	2.04	17.78~25.35	6.75	0.10	−0.25				
		17XX	-	-	20.10 ± 0.10	2.31	16.14~23.53	7.56	−0.28	−0.48				
		18XX	-	-	19.20 ± 0.12	2.63	15.29~22.93	8.44	−0.03	−0.54				

^a^ 16SY, 17SY, 18SY, 17XT, 17XX, 18XX—different environments of Sanya, Xiangtan, Xinxiang in 2016, 2017, and 2018. ^b^ HH27—Heihe27; ZGDD—Zigongdongdou. ^c^ CV—coefficient of variation. ^d^
*H*^2^—broad-sense heritability. ^e^ NA—not available. ^f^ ** *p* < 0.01; *** *p* < 0.001. ^g^ RILs—recombinant inbred lines.

**Table 2 ijms-20-05915-t002:** Co-detected QTL identified by linkage analysis in two-algorithm and/or multiple growth environments.

Trait	QTL Name	Chr. (LG) ^a^	Method ^b^	Location (cM)	Marker Interval (cM)	Physical Region (bp)	LOD/F Value ^c^	PVE (%) ^d^	Additive Effect ^e^	Environment ^f^	Reference ^g^
Protein	*qPro1-1*	1 (D1a)	ICIM	1	0~1.5	1488983~1566969	2.90~6.49	2.73~5.52	−0.41~−0.49	1, 3	Seed protein 3-4
			MCIM	7.5	6.5~8.4	2605140~2852655	5.00	-	−0.24	-	
	*qPro6-1*	6 (C2)	ICIM	33	32.5~33.5	5836780~5931027	4.93	4.17	0.42	1	cqSeed protein-005, Seed protein 30-5
			MCIM	32.1	31.7~32.2	5609477~5632020	5.10	-	0.26	-	
	*qPro9-1*	9 (K)	ICIM	62~68	59.5 ~70.5	38117239~41020511	3.63~8.96	3.43~8.40	0.38~0.41	1, 2, 4	Seed protein 33-3, Seed protein 34-6
			MCIM	61.7	60.7~62.7	38117239~39894925	10.40	-	0.42	-	
	*qPro12-1*	12 (H)	ICIM	105~107	103.5~107	38776571~39867556	3.64~3.78	3.05~3.58	0.36~0.46	1, 3	Seed protein 6-1
	*qPro15-1*	15 (E)	ICIM	23~26	22.5~26.5	2691560~3476238	9.00~19.05	13.40~17.81	0.79~0.89	3, 4, 5	Seed protein 30-3
			MCIM	26.2	26.1~26.3	3311604~3350307	26.20	-	0.52	-	
	*qPro18-1*	18 (G)	ICIM	22	21.5~22.5	5577815~5618246	6.06	5.80	−0.59	3	Seed protein 47-6
			MCIM	22.3	22.0~23.3	5618246~5979842	4.80	-	−0.24	-	
	*qPro20-1*	20 (I)	ICIM	54~61	48.5~62.5	34734798~37115770	6.14~8.62	7.24~9.39	0.56~0.75	1, 2, 3	Seed protein 26-5, Seed protein 34-11
			MCIM	58.7	57.7~59.7	36089907~37115770	15.30	-	0.34	-	
Oil	*qOil1-1*	1 (D1a)	ICIM	1~10	0~14.5	1488983~3316074	2.85~2.99	1.56~1.67	0.14~0.17	1, 3	Seed oil 23-2
	*qOil2-1*	2 (D1b)	ICIM	121	116.5~126.5	43783867~45442501	2.56	3.64	0.19	5	cqSeed oil-014, Seed oil 39-6
			MCIM	112.8	111.8~113.1	42545649~43226016	5.40	-	0.14	-	
	*qOil3-1*	3 (N)	ICIM	52~54	50.5 ~55.5	33430615~34447425	2.62~2.72	1.86~2.78	0.15~0.21	2, 4	Seed oil 43-30
	*qOil5-1*	5 (A1)	ICIM	117~126	116.5~126	40003403~41813079	3.89~35.35	7.04~23.98	−0.27~−0.63	1, 2, 3, 4, 5, 6	Seed oil 39-1, Seed oil 35-2, Seed oil 13-1
			MCIM	125.9	124.9~126.4	40566361~41813079	25.10	-	−0.40	-	
	*qOil6-1*	6 (C2)	ICIM	50~52	44.5~52.5	8313637~9652882	3.24~4.09	4.16~6.68	−0.26~-0.34	2, 6	cqSeed oil-016
	*qOil15-1*	15 (E)	ICIM	26	25.5~26.5	2691560~3240013	19.25	15.97	−0.44	4	cqSeed oil-007, Seed oil 2-3
			MCIM	26.2	26.1~26.3	3311604~3350307	28.40	-	−0.28	-	
	*qOil17-1*	17 (D2)	ICIM	41	39.5~41.5	7100839~8674575	3.19	1.72	0.18	3	Seed oil 23-3
			MCIM	45.1	44.1~46.1	7453724~9120650	5.30	-	0.13	-	
	*qOil20-1*	20 (I)	ICIM	56~62	51.5~62.5	34734798~37115770	3.93~5.20	2.30~2.87	−0.20	1, 3	Seed oil 27-4, Seed oil 24-6

^a^ Chr. (LG), chromosome (linkage group). ^b^ inclusive composite interval mapping (ICIM) and a mixed model based on composite interval mapping (MCIM) were used. ^c^ limit of detection (LOD) value was the threshold by ICIM, and *F*-value was the threshold by MCIM, respectively, with the critical threshold value LOD = 2.5 and *F* = 4.7, respectively. ^d^ PVE, explanation of phenotypic variation. ^e^ Positive value means the ZGDD allele contributed to the trait. ^f^ 1, 2, 3, 4, 5, 6 represented 16SY, 17SY, 18SY, 17XT, 17XX, 18XX, respectively. ^g^ Reported quantitative trait loci (QTL) in Soybase databse (https://www.soybase.org/) that overlapped our QTL here.

**Table 3 ijms-20-05915-t003:** Co-detected SNP loci regions by linkage analysis and GWAS.

Chr. ^a^	Trait	Method ^b^	Environment ^c^	Markers Interval (cM)/SNP Number ^d^	SNP Loci Region/Location (bp)	LOD/F Value ^e^	PVE (%) ^f^	Additive Effect ^g^
2	Oil	ICIM	5	116.5~126.5	43783867~45442501	2.56	3.64	0.19
		GWAS	4	1	45017225	-	-	-
6	Protein	ICIM	1	32.5~33.5	5836780~5931027	4.93	4.17	0.42
		MCIM	-	31.7~32.2	5609477~5632020	5.10	-	0.26
	Oil	GWAS	3, 5	2	5713084~5992538	-	-	-
9	Protein	ICIM	1, 2, 4	59.5~70.5	38117239~41020511	3.63~8.96	3.43~8.40	0.38~0.41
		MCIM	-	60.7~62.7	38117239~39894925	10.40	-	0.42
	Oil	GWAS	6	1	40301013	-	-	-
20	Protein	ICIM	1, 2, 3	48.5~62.5	34734798~37115770	6.14~8.62	7.24~9.39	0.56~0.75
		MCIM	-	57.7~59.7	36089907~37115770	15.30	-	0.34
		GWAS	5, 6	5	34990940~35578946	-	-	-
	Oil	ICIM	1, 3	51.5~62.5	34734798~37115770	3.93~5.20	2.30~2.87	−0.20
		GWAS	5	4	34801441~35512580	-	-	-

^a^ Chr. chromosome. ^b^ Inclusive composite interval mapping (ICIM) and a mixed model based on composite interval mapping (MCIM) were two algorithms in linkage analysis. ^c^ 1, 2, 3, 4, 5, 6 represented 16SY, 17SY, 18SY, 17XT, 17XX, 18XX, respectively. ^d^ Markers interval is the QTL interval in linkage analysis, SNP number is the significant SNP loci number in the SNP loci region. ^e^ LOD value is the threshold by ICIM and *F*-value is the threshold by MCIM, respectively, with the critical threshold value LOD = 2.5 and *F* = 4.7, respectively. ^f^ PVE, explanation of phenotypic variation. ^g^ Positive value means ZGDD allele contributed to the trait.

**Table 4 ijms-20-05915-t004:** Candidate genes that may control protein/oil content within the SNP regions on Chr. 15 and 20.

Trait	Gene	Start (bp)	Stop (bp)	Annotation
Oil	*Glyma.15g034100*	2722009	2727957	acyltransferase activity, diacylglycerol and triacylglycerol biosynthesis
	*Glyma.15g034400*	2740960	2746344	aldehyde dehydrogenase family, aldehyde dehydrogenase [NAD (P)+] activity
	*Glyma.15g034600*	2765299	2770528	drug transmembrane transport, associated with the transport of citric acid and malic acid
	*Glyma.15g042500*	3339156	3341447	fatty acid, lipid biosynthetic process, transferase activity, 3-oxoacyl-[acyl-carrier-protein] synthase activity
	*Glyma.20g107600*	35025241	35029762	Arabidopsis phospholipase-like protein, regulation of gene expression
	*Glyma.20g107800*	35038552	35042864	hydroxypyruvate reductase, glycerate dehydrogenase, glyoxylate reductase, NADP activity
	*Glyma.20g108800*	35116048	35118928	mitochondrial pyruvate transmembrane transport
	*Glyma.20g109900*	35222837	35228540	lipid metabolic process, steroid biosynthetic process, mevalonate pathway
	*Glyma.20g110000*	35229423	35231861	acetyltransferase activity
	*Glyma.20g111000*	35315630	35319063	fatty acid desaturase, lipid metabolic process
Protein	*Glyma.15g033200*	2656030	2657795	structural constituent of ribosome, 28S ribosomal protein
	*Glyma.15g039000*	3068347	3075209	60S ribosomal protein
	*Glyma.15g040100*	3164697	3168839	ACT domain-containing protein, metabolic process like protein synthesis and degradation.
	*Glyma.15g041500*	3255042	3256599	ribosomal large subunit assembly, 60S ribosomal protein L23
	*Glyma.15g042300*	3307111	3308840	structural constituent of ribosome, 60S ribosomal protein L35
	*Glyma.20g103200*	34605252	34609867	tryptophan biosynthetic process, anthranilate synthase activity
	*Glyma.20g105300*	34757381	34771672	ACT-like protein, serine/threonine kinase family protein
	*Glyma.20g106200*	34862155	34865242	amino acid transmembrane transport
	*Glyma.20g106900*	34962043	34967985	translation initiation factor 3 (IF-3) family protein
	*Glyma.20g109600*	35200934	35205885	ubiquitin-dependent protein catabolic process, proteasome complex, proteolysis activity
	*Glyma.20g110100*	35232344	35233758	nutrient reservoir activity, cupins superfamily protein, storage protein
	*Glyma.20g110400*	35261156	35268971	ACT domain-containing protein, metabolic process like protein synthesis and degradation.
	*Glyma.20g111900*	35396205	35400722	cationic amino acid transporter, amino acid transmembrane transporter activity
Protein/Oil	*Glyma.20g106800*	34935548	34940516	protein dephosphorylation, phosphatase activity, pyruvate dehydrogenase
	*Glyma.20g110200*	35235204	35239070	lipoate biosynthetic, radical SAM superfamily protein, transferase activity

ACT—Aspartate kinase, Chorismate mutase and TyrA (prephenate dehydrogenase), SAM—S-adenosyl methionine, NADP—nicotinamide adenine dinucleotide phosphate, IF—initiation factor.

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
