# Peer review of "A Combined Linkage and GWAS Analysis Identifies QTLs Linked to Soybean Seed Protein and Oil Content"

_ijms, 2019, doi:10.3390/ijms20235915_

Round 1

Reviewer 1 Report

Authors presented manuscript regarding GWAS and QTL ananlyses from 308 RIL (RAD-seq) and 203 WGS data. However, many important information is not provided.

Only names and origins of 203 accession are provided without any information about WGS data. Usually, it is required to upload and open WGS data publicly. Bioinformatics methods how they analyze those data are described very shortly and it need to be improved.

qq-plot results from GWAS shows their are population bias because blue circles are above read line near zero point. Many people remove population bias based on PCA results.

Candidate genes are in Chr15 and Chr20, but Chr15 is not co-detected. Please justify.

There have been many researches on protein and oil contents. Please describe novel results and replicate results in previous results. One may find QTL results in soybase.org

Figure 3. LD decay can be caculated for four populations; NER, HHR, SR, and USA. fastStructure can be done for other K values, 3, 4, 5 because they have at least 4 populations.

Table 5. "Trait" in this table need to be modified. It looks like those genes are regulating protein and oil contents. This table need to be separated according to two traits, and describe general gene information including functions, GO, pathway, plant ontology and etc.

Author Response

Response to reviewer 1

Point 1: Authors presented manuscript regarding GWAS and QTL ananlyses from 308 RIL (RAD-seq) and 203 WGS data. However, many important information is not provided.

Response 1: According to your valuable suggestions, the authors have revised our manuscript thoroughly. The authors have adjusted and reconducted PCA, population structure and GWAS, and compared our result with previous studies. The details were revised in the manuscript.

Point 2: Only names and origins of 203 accession are provided without any information about WGS data. Usually, it is required to upload and open WGS data publicly. Bioinformatics methods how they analyze those data are described very shortly and it need to be improved.

Response 2: Datasets of 134 accessions supporting the results of this article are available in National Center for Biotechnology Information (NCBI) repository. Short re-sequencing reads were deposited at Short Read Archive (SRA) under accession number of SRP062560. Identified SNPs were deposited at dbSNP under accession number of SRP062560 and datasets of the rest 69 accession were uploading now, and it will finish in the next few days.

The authors added the bioinformatic analysis of WGS data in “4.4. Genotyping and GWAS” in line 357-360 as “The 203 soybean cultivars were sequenced with the high-throughput next-generation sequencing platform of Hi-Seq 2000 with an average sequencing depth of 10 fold and genotyped using the GATK pipeline. After being trimmed by TRIMMOMATIC (parameter: ILLUMINACLIP: adaptor.seq:2:30:10 TRAILING:3 SLIDINGWINDOW:4:10 MINLEN:20), clean reads were mapped to soybean reference genome (version Wm82.a2.v1)by BWA mem47 with default parameters, SNPs/InDels were called by GATK (-stand_call_conf set to 30.0, -stand_emit_conf set to 10.0, and -glm set to BOTH). The variations were then recalibrated by Gaussian mixture model, and outliers were discarded.Variants were further filtered by BCFtools (version 1.2, QUAL ≥ 50.0, DP ≥ 5.0, QD ≥ 5.0, MQ ≥ 30, MAF≥ 0.03, Coverage≥ 90%). InDels longer than 6 bp were discarded.” 

Point 3: qq-plot results from GWAS shows their are population bias because blue circles are above read line near zero point. Many people remove population bias based on PCA results.

Response 3: Thanks for your valuable suggestion. The authors have conducted PCA on the LD pruning SNPs to detect the population bias. 3 cultivars (Hai 94, Wuhuasiyuehuang and Suidaohuang) of population bias was identified and removed based on the PCA result. Besides, to minimize false positives due to population structure, mixed linear model (MLM) was conducted to reduce false positive SNPs on two sub-population panels (NER-USA sub population and HHR-SR sub-population) as well as total association panel. The QQ-plots indicated our results are highly improved and convinced. A total of 19, 12, 36 SNP loci distributed on 17 chromosomes were detected in NER-USA sub population, HHR-SR sub-population and total population. The P values of all significant SNP loci were from 9.37×10-7 to 9.90×10-5. One SNP on Chr. 1, 5, 9, 13, 16, 18 and two SNPs on Chr. 20 were associated with both trait in one environment, and the significant SNP loci on Chr. 20 were associated with both protein and oil content in 17XX and 18XX. The PCA plots, manhanttan plots and QQ-plots were as follows.

 Before delete outlier:

After delete outlier:

Manhanttan plots and QQ-plots in subgroup1 (NER-USA):

Manhanttan plots and QQ-plots in subgroup2 (HHR-SR):

Manhanttan plots and QQ-plots in total association panel:

The authors replace our previous GLM results with MLM now in “2.3. GWAS results” in line 145-177, Figure 3 and “Supplementary file: Table S1 and Figure S4”.

Point 4: Candidate genes are in Chr15 and Chr20, but Chr15 is not co-detected. Please justify.

Response 4: The QTL interval on Chr. 15 was not co-detected by GWAS. SNP regions for candidate genes exploring here were those had high phenotypic variation explanation (PVE) and stable reproducibility in multiple environments. Among the co-detected results, SNP regions on 20 were satisfactory with the above requirement. Although the interval on Chr. 15 was not co-detected, it’s high PVE for both traits, stable reproducibility and narrow physical region do interest us, it’s an extra region to explore candidate genes.

The authors add the figure of LOD value and F value for qPro15-1/qOil15-1, qPro20-1/qOil20-1 and qOil5-1 in “Supplement file: Figure S1” and in “Result2.2” in line 134-135.

The authors change our statement of these two regions as “the co-detected SNP loci region 34.60~35.40 Mb extended by the LD decay distance of 200 kb on Chr. 20, and the qPro15-1/qOil15-1 interval 2.60~3.50 Mb on Chr. 15, an extra region, were the best novel regions for candidate genes, rather novel QTL intervals, that might control protein and/or oil metabolism.” in “Discussion3.3” in line 281-284, and change the description of the qPro15-1/qOil15-1 region in “Result2.5” in 194-198.

Point 5: There have been many researches on protein and oil contents. Please describe novel results and replicate results in previous results. One may find QTL results in soybase.org

Response 5: The authors are sorry to reply to you that we made an inappropriate statement in the title. The word “novel” was not suitable for our QTL results, all of our QTL intervals overlapped or closed to regions reported by previous studies and we listed previous QTL overlapping with ours in Table 2.

The authors revise the title as "Combined Linkage Analysis and GWAS Method Reveals Refined QTL and Novel Candidate Genes Linked to Soybean Seed Protein and Oil Content". We delete the word “novel” in “Abstract”.

Point 6: Figure 3. LD decay can be caculated for four populations; NER, HHR, SR, and USA. fastStructure can be done for other K values, 3, 4, 5 because they have at least 4 populations.

Response 6: According to your valuable suggestion, the authors have calculated the LD decay values and added the LD decay plots of the four populations (Supplementary file: Figure S3) as well as total association panel (Figure 3e). The LD decays were 180 kb, 190 kb, 171kb, 161kb and 132 kb for NER, USA, HHR, SR and total population, respectively. The authors also added the population structure analysis of K values 2, 3, 4, 5 in “Supplementary file: Figure S2”and  decided K=2 is the best model choice.

LD decay of accessions from NER, USA, HHR and SR:

the population structure of K=2, 3, 4, 5 by fastStructure:

Point 7: Table 5. "Trait" in this table need to be modified. It looks like those genes are regulating protein and oil contents. This table need to be separated according to two traits, and describe general gene information including functions, GO, pathway, plant ontology and etc.

Response 7: Thanks a lot for your valuable suggestion. We change the Table 5 format according to two traits. The annotation column in Table 5 was the information that might link to metabolism of protein/oil we extracted from relevant databases such as TAIR, GO, PFAM, PANTHER and KOG

Reviewer 2 Report

This manuscript identified several candidate genes associated with protein and oil contents using bi-parental genetic mapping and GWAS analysis. They found good candidate genes and these results will be useful information for genetic studies and soybean breeding. With some modification, it can be acceptable.

1. First, the "novel" in the title was not match with the results. I don't know which SNPs were novel loci in this study. Also, they focused ch15 and ch20 that identified previously reports. Need to be changed.

2.  In the RT-PCR analysis, the plants were grown different conditions, such as "ZGDD planted in a short-day greenhouse (12 hours light/12 hours dark) and HH27 planted 368 in a long-day greenhouse (16 hours light/8 hours dark) to simulate their native growth conditions". I don't understand why these two parents should be grown at different environments and the gene expression can be compared between two parents.

3. Q-Q plots of figure s1 were not normal Q-Q plots in GWAS analysis. There were too much gap between the normal lines and samples. Should be adjust the analysis methods.

4. In 17XT conditions, the protein content of two parents was not different. Need the explanation about these results. 

Author Response

Response to reviewer 2

This manuscript identified several candidate genes associated with protein and oil contents using bi-parental genetic mapping and GWAS analysis. They found good candidate genes and these results will be useful information for genetic studies and soybean breeding. With some modification, it can be acceptable.

Point 1: First, the "novel" in the title was not match with the results. I don't know which SNPs were novel loci in this study. Also, they focused ch15 and ch20 that identified previously reports. Need to be changed.

Response 1: We agree with your comments. Actually, all of our QTL intervals overlapped or closed to regions reported by previous studies. What we want to state in title is that these QTL in our study are refined QTL with narrow physical regions, high effectiveness and reproducibility. There are no relevant candidates genes have been reported within these regions, so these QTL are better regions to explore genes, not novel QTL regions that don’t have overlaps with previous QTL.

The authors have a deviation in the statement of the original title, now we revise the title as "Combined Linkage Analysis and GWAS Method Reveals Refined QTL and Novel Candidate Genes Linked to Soybean Seed Protein and Oil Content". The authors delete the word “novel” in “Abstract”.

Point 2: In the RT-PCR analysis, the plants were grown different conditions, such as "ZGDD planted in a short-day greenhouse (12 hours light/12 hours dark) and HH27 planted 368 in a long-day greenhouse (16 hours light/8 hours dark) to simulate their native growth conditions". I don't understand why these two parents should be grown at different environments and the gene expression can be compared between two parents.

Response 2: The authors agreed with the reviewer’s valuable suggestion. Indeed, the gene expression comparison should be conducted in the same environmental condition. However, the parent ZGDD originates from the low latitude area of China (Zigong, Sichuan province, N29°20, E104°46′), it is sensitive to photoperiod and can only blossom and mature in a short-day condition. Initially, in order to get the protein and oil content closing to the origin region of each parent cultivar, we simulated the suitable light conditions for both parents. The parent ZGDD originates from the low latitude area of China (Zigong, Sichuan province, N29°20, E104°46′), it is sensitive to photoperiod and can only blossom and mature in a short-day condition; the parent HH27 derives from the high latitude area of China (Heihe, Heilongjiang province, N50°14′, E127°31′), it is insensitive to photoperiod and can blossom and mature in both long-day and short-day conditions. However, the environmental conditions have a great influence on protein and oil content, different light conditions may also affect gene expression and ultimately lead to the different accumulation of protein and oil content. Therefore, in subsequent experiments, it is necessary to further analyze the expression of all candidate genes in different light conditions and different varieties to determine whether gene expression and the accumulation of protein and oil content are related to light condition or the genotype.

The authors added these descriptions in the “Materials and Methods4.5” in line 382-388, as well as the “Discussion3.3” in line 312-316.

Point 3: Q-Q plots of figure s1 were not normal Q-Q plots in GWAS analysis. There were too much gap between the normal lines and samples. Should be adjust the analysis methods.

Response 3: Thanks for your valuable suggestion. Thanks for your valuable suggestion. The authors have conducted PCA on the LD pruning SNPs to detect the population bias. 3 cultivars (Hai 94, Wuhuasiyuehuang and Suidaohuang) of population bias was identified and removed based on the PCA result. Besides, to minimize false positives due to population structure, mixed linear model (MLM) was conducted to reduce false positive SNPs on two sub-population panels (NER-USA sub population and HHR-SR sub-population) as well as total association panel. The QQ-plots indicated our results are highly improved and convinced. A total of 19, 12, 36 SNP loci distributed on 17 chromosomes were detected in NER-USA sub population, HHR-SR sub-population and total population. The P values of all significant SNP loci were from 9.37×10-7 to 9.90×10-5. One SNP on Chr. 1, 5, 9, 13, 16, 18 and two SNPs on Chr. 20 were associated with both trait in one environment, and the significant SNP loci on Chr. 20 were associated with both protein and oil content in 17XX and 18XX. The PCA plots, manhanttan plots and QQ-plots were as follows.

Before delete outlier:

After delete outlier:

Manhanttan plots and QQ-plots in subgroup1 (NER-USA):

Manhanttan plots and QQ-plots in subgroup2 (HHR-SR):

Manhanttan plots and QQ-plots in total association panel:

The authors replace our previous GLM results with MLM now in “2.3. GWAS results” in line 145-177, Figure 3 and “Supplementary file: Table S1 and Figure S4”.

Point 4: In 17XT conditions, the protein content of two parents was not different. Need the explanation about these results. 

Response 4: It is a good suggestion. During the field trials, we set two parents control for every 60 lines; all the parents in 17XT were identified by 54 pairs of SSR markers screening, with HH27 and ZGDD as the positive control, and it was found that the lines were not mixed. Therefore, we believe the comprehensive climate factors in XT lead to a significant change in protein and oil content of HH27.

The author add our inference “the comprehensive climate factors in XT must lead to the significant change in protein and oil content of HH27” in the “Discussion3.1” in line 232-233.

Round 2

Reviewer 1 Report

Authors made significant changes and improvements over first draft. Here are comments.

In authors reply, they said they are uploading NGS reads, but SRP062560 is not described in main text. Please check.

LD decay values are not described in main text, only in authors reply. Please check. Again, I think smoothening algorithm is not applied in LD decay plot. It may give better plot if LOWESS algorithm is applied.

There is no table 5. I guess table numbers are changed? Please check again.

Figure S2. Please sort the fastSTRUCTURE results and modify colors to see changes. It looks like everything is mixed up.

PCA results in main text: dots are too small to see. Please enhance PCA results. Again, fastSTRUCTURE results in Figure 3 can be sorted (accessions can be sorted) to show better picture.

Author Response

Authors made significant changes and improvements over first draft. Here are comments.

Point 1: In authors reply, they said they are uploading NGS reads, but SRP062560 is not described in main text. Please check.

Response 1: Thanks so much for your valuable suggestion. The authors have added “The sequencing data of 125 accessions used in this study have been deposited into the NCBI database under Short Read Archive (SRA) accession number SRP062560, and the sequencing data of the rest 78 accessions used only in this study are depositing into SRA database in NCBI under accession number PRJNA589345.” in the main text in line 367-371.

Point 2: LD decay values are not described in main text, only in authors reply. Please check. Again, I think smoothening algorithm is not applied in LD decay plot. It may give better plot if LOWESS algorithm is applied.

Response 2: Thanks for your valuable suggestions. The LD decay values of the four populations as well as the total association panel were showed in line 160-163 as“The physical distance of total LD decay, where r2 dropped below 0.1, was approximately 132 kb (Figure 3e), and we also detected the LD decay of accessions from NER, USA, HHR and SR with the relative LD decay distances of 180 kb, 190 kb, 171kb and 161kb, respectively.”in the revised manuscript.

The authors also apply LOWESS algorithm to modify the LD plots and add “LD plots were modified via locally weighted scatterplot smoothing (LOWESS) using R software and testing smoothing parameters fixed to 0.01.” in line 373-374. The modified figure is shown in Figure 3e and also shown in the following figure from the file attached.

Point 3: There is no table 5. I guess table numbers are changed? Please check again.

Response 3: Thanks so much for your comments. The authors have changed the table numbers. The previous Table 5 was changed to Table 4 in the revised manuscript. We changed the Table 4 format according to your previous suggestion by two traits. The annotation column in Table 4 was the information that might link to metabolism of protein/oil we extracted from relevant databases such as TAIR, GO, PFAM, PANTHER and KOG.

Point 4: Figure S2. Please sort the fastSTRUCTURE results and modify colors to see changes. It looks like everything is mixed up.

Response 4: Thanks for your valuable suggestion. We modified the Figure S2 into a more suitable format to see the structure differences between different K value clearly. The revised figure is also shown in the file attached.

Point 5: PCA results in main text: dots are too small to see. Please enhance PCA results. Again, fastSTRUCTURE results in Figure 3 can be sorted (accessions can be sorted) to show better picture.

Response 5: Thanks for your valuable recommendation. The authors have revised the PCA plot in Figure 3b which can show larger dots in. The authors have sorted the accessions and shown better pattern in Figure 3c. The revised Figure 3 is shown in the file attached, too.
